# Investigating hospital *Mycobacterium chelonae* infection using whole genome sequencing and hybrid assembly

**Christopher H. Gu**[1], **Chunyu Zhao**[2], **Casey Hofstaedter**[2], **Pablo Tebas**[3], **Laurel Glaser**[4], **Robert Baldassano**[2], **Kyle Bittinger**[2]*, **Lisa M. Mattei**[2]*, **Frederic D. Bushman**[1]*

**1** Department of Microbiology, University of Pennsylvania School of Medicine, Philadelphia, Pennsylvania, United States of America, **2** Division of Gastroenterology, Hepatology, and Nutrition, Children's Hospital of Philadelphia, Philadelphia, Pennsylvania, United States of America, **3** Department of Medicine, University of Pennsylvania School of Medicine, Philadelphia, Pennsylvania, United States of America, **4** Department of Pathology and Laboratory Medicine, University of Pennsylvania School of Medicine, Philadelphia, Pennsylvania, United States of America

☯ These authors contributed equally to this work.
* bushman@pennmedicine.upenn.edu (FDB); matteil@email.chop.edu (LMM); bittingerk@email.chop.edu (KB)

**Data Availability Statement:** Nanoflow is available at www.github.com/zhaoc1/nanoflow while all other computer code used in this study is available at www.github.com/chrgu. M. chelonae and V.

## Abstract

*Mycobacterium chelonae* is a rapidly growing nontuberculous mycobacterium that is a common cause of nosocomial infections. Here we describe investigation of a possible nosocomial transmission of *M. chelonae* at the Hospital of the University of Pennsylvania (HUP). *M. chelonae* strains with similar high-level antibiotic resistance patterns were isolated from two patients who developed post-operative infections at HUP in 2017, suggesting a possible point source infection. The isolates, along with other clinical isolates from other patients, were sequenced using the Illumina and Oxford Nanopore technologies. The resulting short and long reads were hybrid assembled into draft genomes. The genomes were compared by quantifying single nucleotide variants in the core genome and assessed using a control dataset to quantify error rates in comparisons of identical genomes. We show that all *M. chelonae* isolates tested were highly dissimilar, as indicated by high pairwise SNV values, consistent with environmental acquisition and not a nosocomial point source. Our control dataset determined a threshold for evaluating identity between strains while controlling for sequencing error. Finally, antibiotic resistance genes were predicted for our isolates, and several single nucleotide variants were identified that have the potential to modulated drug resistance.

## Introduction

*Mycobacterium chelonae*, a rapidly growing nontuberculous mycobacterium (NTM), is ubiquitous in the environment and is a common source of opportunistic infection in humans. *M. chelonae* has caused several outbreaks from single point sources, but no human-to-human transmission has been observed to date [1–6]. *M. chelonae* is most commonly associated with

campbellii assembled genomes available at GenBank under project PRJNA594977.

**Funding:** This work was supported by the Penn Center for AIDS Research P30 AI 045008 (https://www.med.upenn.edu/cfar)under FDB; and the PennCHOP Microbiome Program (https://pennchopmicrobiome.chop.edu). The funders had no role in study design, data collection and analysis, decision to publish, or preparation of the manuscript.

**Competing interests:** The authors have declared that no competing interests exist.

nosocomial soft tissue infections of the skin and eye, but can also cause catheter-associated infections, disseminated and invasive infections, and pulmonary infections [1]. *Mycobacterium* can be highly resistant to many antibiotics because of their naturally impermeable cell walls as well as mutation of bacterial genes that encode antibiotic targets [7–9].

Among the *M. chelonae* isolates collected in 2017 from the Hospital of the University of Pennsylvania (HUP) clinical microbiology laboratory were two isolates from female patients who underwent surgery within the same year and developed post-operative surgical site infections in breast tissue. Culture of the inflammatory lesions yielded *M. chelonae* isolates with similar drug resistance patterns, raising suspicion of a nosocomial point source infection. To investigate this possibility, we performed whole genome sequencing on every *M. chelonae* isolate collected by the HUP clinical microbiology laboratory over a one-year period (n = 7) and examined differences in the single nucleotide variants (SNVs) of each isolate's core genome. As a control, we created a dataset from a single *Vibrio campbellii* strain that was sequenced 39 times independently, allowing us to determine a threshold of SNVs that would distinguish different strains while controlling for sequencing error. Finally, we investigated genes and SNVs related to drug resistance in each isolate.

## Methods

Samples were collected as part of routine clinical practice. Penn IRB approved collection of human samples by the clinical Microbiology Laboratory at the Hospital of the University of Pennsylvania under the IRB protocol #829497. Patients had written consent from their providers. The IRB protocol also included a waiver of consent. Respiratory samples were decontaminated with NaOH and N-acetylcysteine. Tissue specimens were pulverized in a tissue grinder. Prepared specimens were inoculated on 7H11 selective and non-selective solid media and a Mycobacterial Growth Indicator Tube (MGIT) broth. All cultures were incubated at 35–37˚C for 6 weeks. Positive cultures for *Mycobacterium* were identified at the species level using hsp65 gene sequencing with primers TB11 (`ACCAACGATGGTGTGTCCAT`) and TB12 (`CTT GTCGAACCGCATACCCT`) [10]. Susceptibility testing and MIC determination were performed using the RAPMYCO microbroth dilution plate (ThermoFisher, catalog number RAPMYCO) and susceptibility was determined using the CLSI M24-A2 [11]. Tigecycline susceptibility breakpoints for the MIC have not been established for *Mycobacterium* [12]. For our analysis, we based our tigecycline thresholds for resistance of ≤0.25 on the methods in Wallace et al. [13]. For antibiotic resistance gene investigation, we categorized susceptibility into susceptible, intermediate, and resistant.

All *Mycobacterium* samples had been previously frozen for routine clinical purposes. Isolates were re-isolated on chocolate agar and propagated by growth in Middlebrook 7H9 media for 5 days. Multiple DNA purification methods were compared to identify one producing high molecular weight DNA in good yield. Ultimately, DNA was purified from each sample using a phenol-chloroform DNA extraction designed for high molecular weight DNA [14]. Long-read libraries were prepared using the Rapid Barcoding Kit, version SQK-RBK004 (Oxford Nanopore, Oxford, UK) and sequenced on the MinION using a R9.4.1 flow cell. Short-read libraries were prepared using the TruSeq DNA Nano Library Prep Kit (Illumina, San Diego, CA), and sequenced on the HiSeq 2500 using 2x125 bp chemistry.

*V. campbellii* was grown in Difco Marine broth 2216 culture media (BD) overnight and DNA was extracted using DNeasy Blood & Tissue Kits (Qiagen). Short read libraries were prepared using the Nextera XT Library Prep Kit (Illumina, San Diego, CA) and sequenced on the HiSeq 2500 using 2x125bp chemistry.

We used the Sunbeam v1.3 [15] pipeline to process the short reads and a custom snakemake v.5.6.0 pipeline, Nanoflow (https://github.com/zhoac1/nanoflow), to process the long reads and perform the hybrid assembly. Short read processing included trimming adapters off reads using Trimmomatic (parameters: leading: 3; trailing: 3; slidingwindow: [4, 15]; minlen:36), filtering out low quality reads using fastqc and removing low complexity reads using Komplexity (parameters: kz_threshold: 0.55) [15]. Long read processing included base calling using Albacore v2.3.4 (https://community.nanoporetech.com), trimming adaptors using Porechop v0.2.3_seqan2.1.1 (https://github.com/rrwick/Porechop), and filtering using Filtlong v0.2.0 (parameters:—min_length 1000—keep_percent 90—target_bases 1000000000) (https://github.com/rrwick/Filtlong). Hybrid assembly was performed by two methods: 1) using Canu v1.9, polishing the genome using Nanopolish v0.11.1 and correction with short reads using Pilon or 2) using Unicycler v0.4.7, a program that uses the short read assembler Spades v3.12 guided by long reads as scaffolds, which is further polished using Pilon [16, 17]. CheckM v1.1.2 and alignment to reference genome (*M. chelonae* strain CCUG 47445) were used to check the quality of the draft genomes to select the best one [18].

We used another custom snakemake pipeline, CoreSNPs (https://github.com/chrgu/coreSNPs) to investigate how related the clinical isolates were to each other and to other whole genome sequenced references collected from GenBank. CoreSNPs uses Prokka v1.14.5 for genome annotation, and Roary v3.12.0 for investigating the pangenome and a hierarchical cluster based on the presence and absence of accessory genes [19, 20]. Custom R and shell scripts were used to extract the core genes from the isolates to compare SNVs by hamming distance. SNVs per Mbp core genome were calculated dividing the hamming distances by length of core genome between the sequence sets for identical DNAs. The SNV analysis used SamTools v1.9 and SNP-sites v2.4.1 [21, 22]. Approximately maximum-likelihood phylogenetic trees for the core genes were generated by Fasttree 2 v2.1.10 [23]. Accession numbers for all *M. chelonae* isolates and references can be found in S1 Table.

The threshold for calling strain identity between genomes was determined using a dataset of a single isolate of *V. campbellii* sequenced 39 times (Illumina only) and examining the number of SNVs between sequencing runs. The dataset was analyzed using the same in-house pipelines as above. Roary defined core genes as genes present in 100% of the samples.

We used Resistance Gene Identifier (RGI) on the output of Prokka to examine resistance genes within the genome [24]. We searched for both high identity and low identity homologous hits to identify previously known genes, such as *gyrA*, and potentially novel resistance genes, respectively. We also manually aligned known genes that can harbor resistance mutations with Muscle to identify and compare mutations within the gene that corresponded with drug susceptibility [25].

Nanoflow is available at www.github.com/zhaoc1/nanflow while all other computer code used in this study is available at www.github.com/chrgu. *M. chelonae* and *V. campbellii* assembled genomes are available in GenBank under project PRJNA594977.

## Results

*M. chelonae* was isolated from seven patients at the Hospital of the University of Pennsylvania in 2017 (Table 1), including the two breast tissue cases. Ages ranged from 45 to 64. Sites of infection included skin (n = 2), breast tissue (n = 2), and respiratory tract (sampled as bronchial alveolar lavage; n = 1) and sputum (n = 2).

Drug susceptibility testing was performed on each isolate using 11 different antibiotics to determine the minimum inhibitory concentration (MIC) (Table 2). As expected, most of the isolates of *M. chelonae* were highly resistant to antibiotics. All strains (n = 7) were either

**Table 1. Patient demographics and the suspected transmission pair.**

| ID | SUSPECTED NOSOCOMIAL INFECTION PAIR* | AGE | SEX | SITE OF COLLECTION |
|---|---|---|---|---|
| MYCO1 | X | 45 | Female | breast |
| MYCO2 | X | 47 | Female | breast |
| MYCO3 | | 61 | Female | skin |
| MYCO4 | | 64 | Male | bronchiolar lavage |
| MYCO5 | | 61 | Male | sputum |
| MYCO6 | | 61 | Male | sputum |
| MYCO7 | | 55 | Male | leg skin |

resistant or intermediately resistant to TMP-SMX, ciprofloxacin, imipenem, moxifloxacin, cefoxitin, and minocycline. Only a few drugs such as clarithromycin, tigecycline, and tobramycin, were effective against all the strains. Some variation in susceptibility and resistance was sporadic (e. g. Myco6 alone was sensitive to doxycycline).

The *M. chelonae* strains were isolated from seven patients at HUP during routine clinical treatment. Frozen stocks of the *M. chelonae* isolates were cultured and DNA was extracted. Extensive optimization was required to allow lysis of the tough *Mycobacterium* cell wall while preserving long DNA chains (see methods). We purified high molecular weight DNA for most strains but we were unable to do so for Myco5. DNA sequencing data was acquired using the Illumina HiSeq 2500 to generate short reads and the Oxford Nanopore MinION to generate long reads. Two assembly methods were compared for each isolate and the best draft genome was chosen based on completeness (by checkM), number and length of contigs, and alignment to a reference genome (described in detail in the methods). For Myco5 only short read assembly was carried out.

**Table 2. Antibiotic resistance profile of *M. chelonae* isolates against 11 antibiotics.**

| ID | TMP* | LINE† | CIPRO§ | IMI¶ | MOXI# | CEFOX** | DOXY†† | MINO§§ | TIGE¶¶ | TOBRA## | CLAR*** |
|---|---|---|---|---|---|---|---|---|---|---|---|
| MYCO1 | >8/152 (R) | 32 (R) | 4 (R) | 16 (I) | 8 (R) | >128 (R) | >16 (R) | 4 (I) | 0.5 (S) | 4 (I) | 0.5 (S) |
| MYCO2A/B | >8/152 (R) | 32 (R) | >4 (R) | >64 (R) | 8 (R) | >128 (R) | >16 (R) | >8 (R) | 0.5 (S) | 2 (S) | 0.5 (S) |
| MYCO3A/B | >8/152 (R) | 32 (R) | >4 (R) | 16 (I) | 8 (R) | 128 (R) | >16 (R) | >8 (R) | 0.5 (S) | 2 (S) | 0.5 (S) |
| MYCO4 | 4/76 (R) | 8 (S) | 4 (R) | 16 (I) | 4 (R) | >128 (R) | >16 (R) | >8 (R) | 0.5 (S) | <1 (S) | 0.25 (S) |
| MYCO5 | 8/152 (R) | 16 (I) | 4 (R) | 32 (R) | 8 (R) | >128 (R) | >16 (R) | >8 (R) | 0.25 (S) | 2 (S) | 0.5 (S) |
| MYCO6 | >8/152 (R) | 16 (I) | 2 (I) | 32 (R) | 4 (R) | 64 (I) | 1 (S) | 2 (I) | 1 (S) | 4 (I) | 0.5 (S) |
| MYCO7 | >8/152 (R) | >32 (R) | 2 (I) | 32 (R) | 4 (R) | >128 (R) | >16 (R) | >8 (R) | 0.5 (S) | 2 (S) | 2 (S) |

Footnote: R = Resistant, I = Intermediate, S = Susceptible

*TMP-SMX

†Linezolid

§Ciprofloxacin

¶Imipenem

#Moxifloxacin

**Cefoxitin

††Doxycycline

§§Minocycline

¶¶Tigecycline

##Tobramycin

***Clarithromycin

In one case, two isolates were cultured from the same patient at different time points and analyzed separately (Myco3a/3b). In another case, a single genomic DNA preparation was sequenced and assembled twice (Myco2a/b). Both pairs provide further empirical data on the sources of error in library preparation and DNA sequencing.

The whole genome sequencing resulted in a range of contig numbers (n = 1 to 76) comprising the main chromosome. For those that were hybrid assembled, the range of contigs was one to four. Three of the nine assemblies yielded complete circular contigs for the main chromosome. The genomes ranged in size from 4.95 to 5.20 Mbp. No clearly defined episomes were found, as judged by detection of extrachromosomal circles (S1 Table).

We assessed the phylogenetic relationships by comparing the number of single nucleotide variants (SNVs) between cores genes (genes found in every isolate), which allowed us to interrogate potential transmission chains. We used all 43 whole genome sequences of *M. chelonae* present in GenBank (retrieved June 2018) as reference to construct a maximum-likelihood phylogenetic tree. The genomes were annotated by Prokka. CheckM analysis were performed to ensure completion and quality of the reference genomes prior to analysis. Analysis of our set of *M. chelonae* genomes returned a total of 17,582 genes in the pan-genome, of which only 3,368 were considered core genes. The length of the total concatenated core genes per genome was 3,296,947 bases. Of the 3,368 core genes, 25 genes did not contain any SNVs. A list of core genes along with the number of SNVs and SNVs per Mbp can be found in S2 Table. There were no obvious genes related with resistance among the top genes with SNVs. Within the core genes, the number of SNVs between unique isolates ranged from 3,383 to 62,854.

Our two samples from the same individual (Myco3a and Myco3b) differed by 3 SNVs while our technical replicates (Myco2a and Myco2b) differed by 2 SNVs. The potential transmission pair, Myco1 and Myco2a/b differed by 16544/16542 SNVs in the core genes (SNVs are indicated for Myco2 replicates a and b, respectively). A maximum likelihood phylogenetic tree based on the SNVs data is shown in Fig 1A. There was no obvious clustering of our clinical isolates compared with database samples. There was some clustering between human samples, and some of our samples fell into those clusters (e. g. Myco3a/3b and Myco5). Other strains clustered with environmental isolates (e. g. Myco2a/2b and Myco4). Likewise, the tree based on presence or absence of accessory genes (Fig 1B) also showed a lack of obvious clustering of the Phildelphia strains.

To assess the likelihood of infection from a common point source, we next empirically assessed the numbers of SNVs expected due to sequencing error in a larger set of genomes. As a positive control in shotgun metagenomic studies, we repeatedly sequenced a single bacterium, *Vibrio campbellii*, a luciferase-encoding marine bacterium, that was divergent from strains likely present in our samples. We recovered an average of 9,072,182 reads over 39 replicates, allowing generation of 39 full genome sequences from the same isolate. Analysis of the *V. campbellii* genomes using Roary disclosed 4495 core genes in our samples; 643 genes, or ~12.5%, were not 100% conserved in the *V. campbellii* genome, likely due to errors in genome sequence determination. Within the core genes, we found a range of SNVs from 0 to 74, with mean of 15 SNVs (Fig 2A). The total length for the concatenated core genes was 4,209,934 bases. The two most divergent *V. campbellii* assemblies also had low sequence coverage (S3 Table). These data provide a rough upper bound on the number of SNV errors associated with suboptimal sequence acquisition.

This comparison takes advantage of Illumina sequence reads only, whereas our *M. chelonae* isolates were sequenced using hybrid assembly of short and long reads. We thus generated short read only assemblies for the *M. chelonae* isolates for a more direct comparison to the *V. campbellii* dataset. Our short-read-only genomes contained slightly fewer core genes (3,143 vs 3,368) compared to our hybrid assemblies. The short-read genomes also had more SNVs per

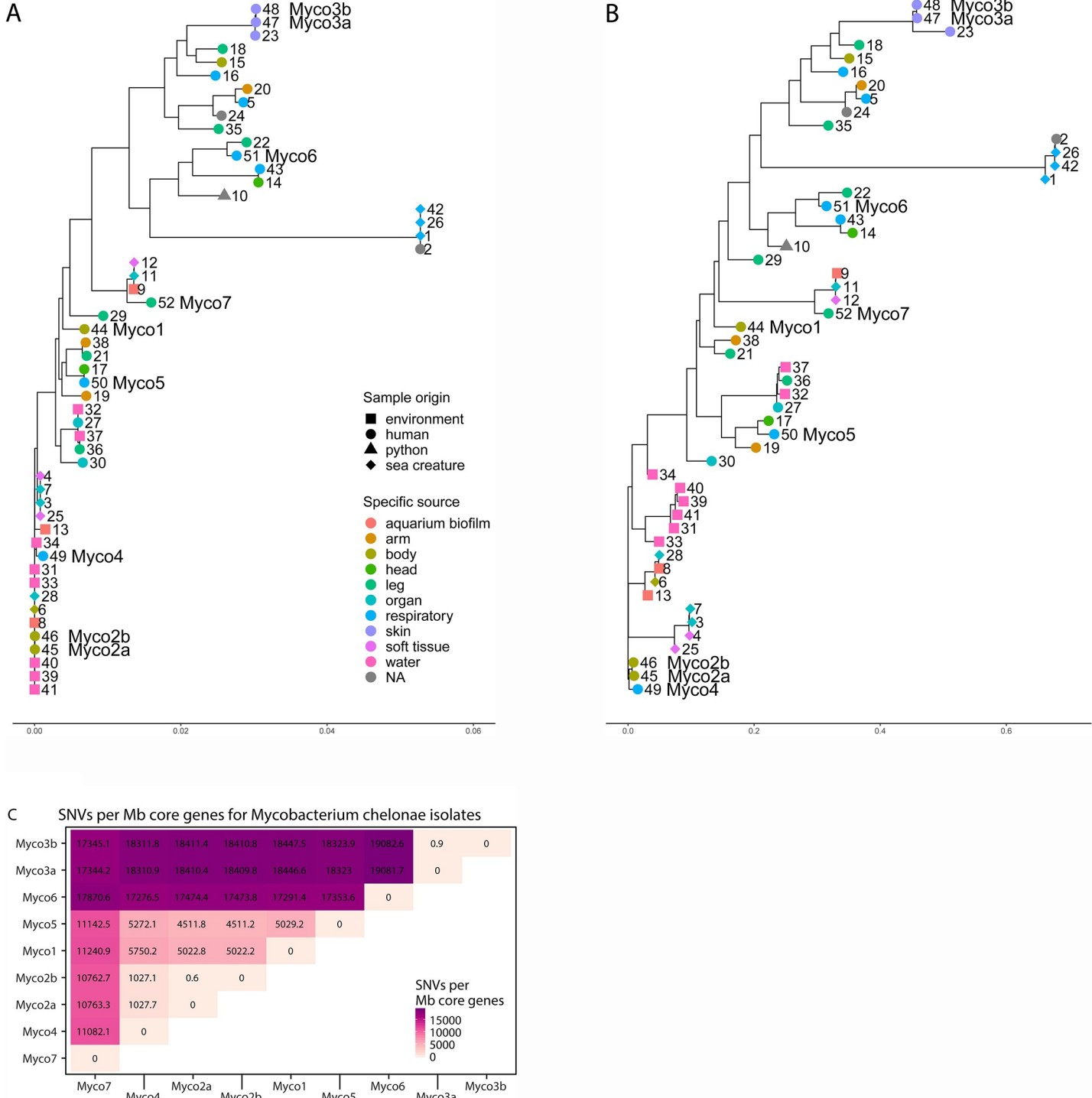

**Fig 1. Relationship of *M. chelonae* genome sequences.** A. A maximum-likelihood phylogenetic tree showing relationships among *M. chelonae* isolates based on SNVs in the core genes. Numbers next to branch tips correspond with genomes found in S1 Table. Isolates from our study are indicated with "Myco" and the isolate number. The sampling site and host of the isolate is coded by color and shape, respectively, at branch tips. The scale at the bottom represents the number of substitutions per sequence site based on length of the tree. B: A maximum-likelihood tree showing the relationship among *M. chelonae* isolates based on presence or absence of accessory genes. C: SNVs per Mbp core genome between *M. chelonae* isolates. SNVs, calculated as hamming distance between the core genes of all isolates divided by the total length of core genes. Myco2a/2b and Myco3a/3b are technical and biological replicates, respectively.

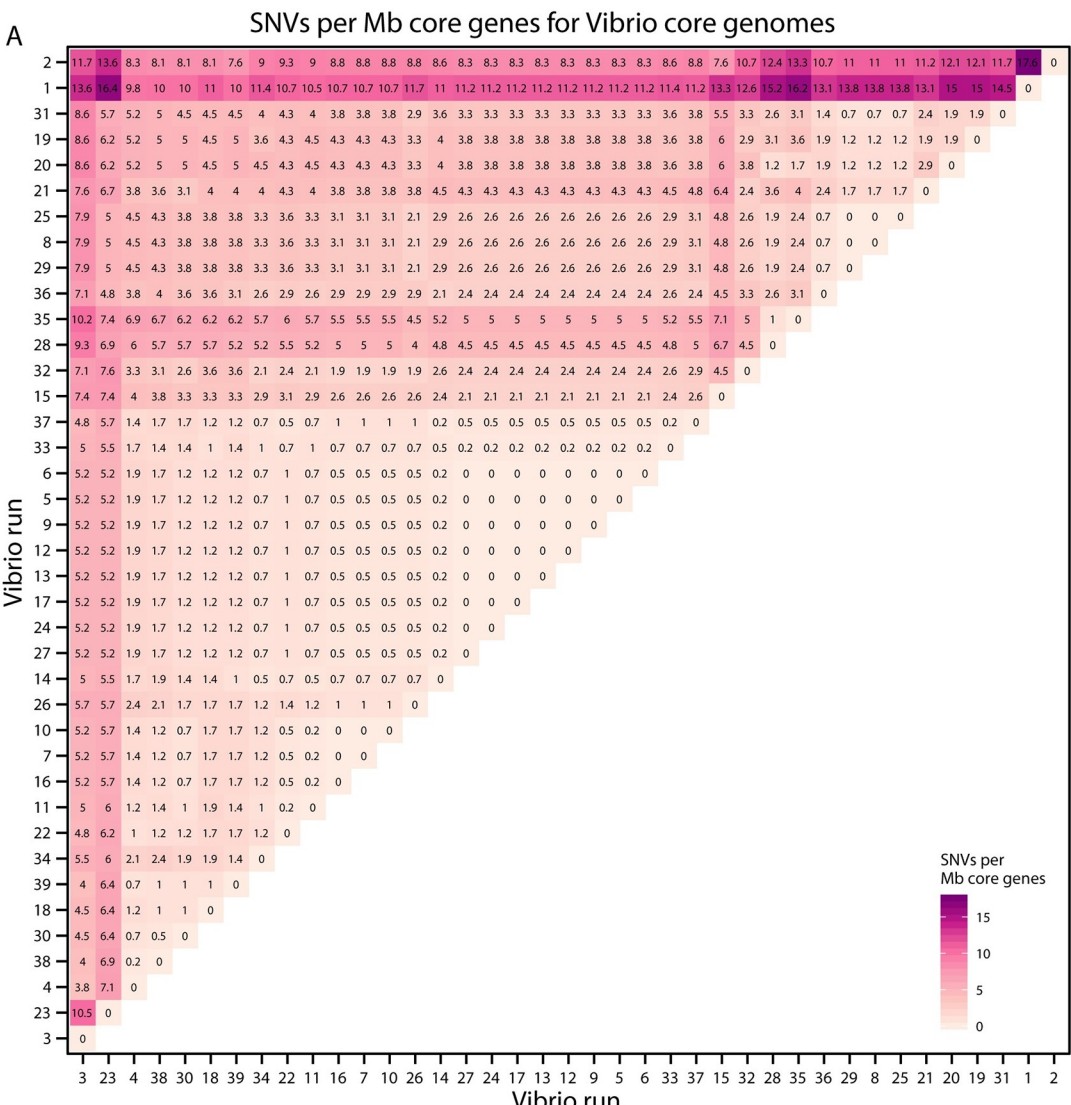

**A** SNVs per Mb core genes for Vibrio core genomes

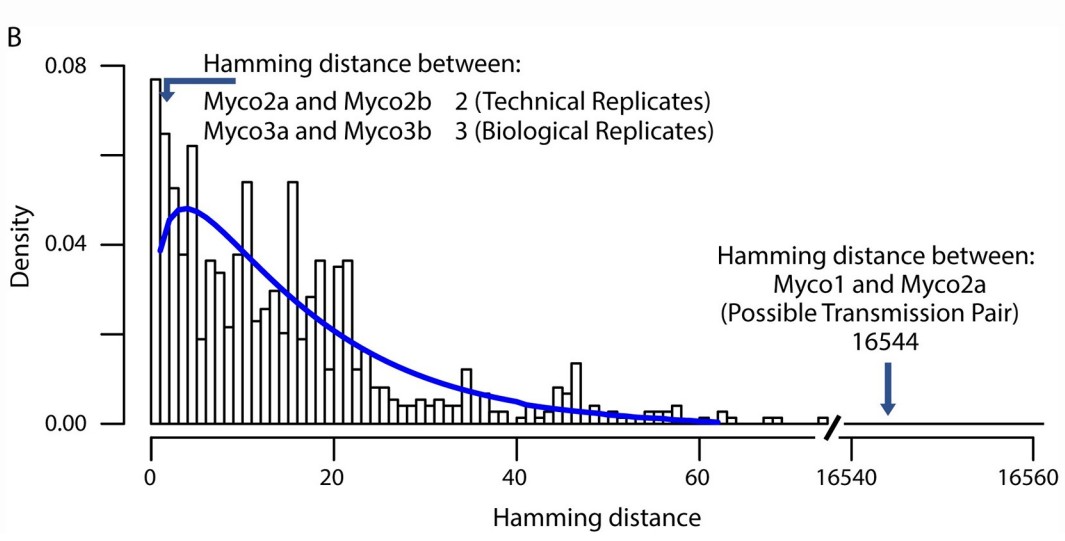

**B**

Hamming distance between:

Myco2a and Myco2b    2 (Technical Replicates)
Myco3a and Myco3b    3 (Biological Replicates)

Hamming distance between:
Myco1 and Myco2a
(Possible Transmission Pair)
16544

**Fig 2. Comparison of *Vibrio campbellii* genomes by SNVs per Mbp core genome to develop statistics for calling isolate identity.** A. The set of SNVs per Mbp core genome, calculated by hamming distances divided by length of core genome between the sequence sets for identical DNAs. B. Graph showing the Hamming distances (x-axis) and the frequencies of distances between pairs (y-axis). The distances between the technical and biological replicates are marked (Myco2a and 2b, and Myco 3a and 3b), as is the distance between the candidate transmission pair (Myco1 and 2a).

Mbp core genes. Both of our technical replicates showed slightly higher SNV counts (3 and 64), but were lower than the maximum number of SNVs for identical strains in our control *V. campbellii* dataset (the maximum number of SNVs for any pair of isolates was 74). Phylogeny of the short-read-only core genomes maintained similar placement as with the hybrid assembled core genome tree.

The possible transmission pair, isolates Myco1 and Myco2a/b (Fig 3), differed by 16,542/16,544 SNVs (Fig 2B), having 99.5% nucleotide identity in the core genes. There were 1107 genes that were not shared between them. Together, this provides strong evidence that they are different strains and not related by direct person-to-person transmission, or acquisition from a common nosocomial point source. This corresponds to a difference of 5,237.58 SNVs per Mbp core genes. The smallest difference between our isolated *M. chelonae* strains was 3,426 SNVs in the core genes or 1,038.18 SNVs per Mbp core genes. Our technical replicates Myco3a and Myco3b differed by 3 SNVs (0.91 SNVs per Mbp core genes), and Myco2a and Myco2b differed by 2 SNVs (0.61 SNVs per Mbp core genes) (Fig 1C). For comparison, the mean number of SNVs in pairwise comparisons of *V. campbelli* control assemblies was 5.71 SNVs per Mbp core genes; the maximum number was 17.62 SNVs per Mbp core genes (Fig 2B). The number of SNVs in the candidate transmission pair thus far exceeds the number of SNVs that could be generated by sequencing error per Mbp as seen from the *V. campbellii* controls and exceeds the SNVs generated in our *M. chelonae* replicates (p-value < 0.001 by binomial test).

The molecular determinants of antibiotic resistance in *M. chelonae* are not well studied, so we sought to assess possible antimicrobial resistance mechanisms disclosed in our sequence data. Analysis using Resistance Gene Identifier (RGI) searching for high identity hits to previously known genes related to resistance [24], yielded two partial hits to all the isolates, and a third in three of seven isolates. The two partial hits in all isolates were *LRA-3* (100% identity) and *erm(38)* (80% identity); the third gene was *mtrA* (95.17% identity). *LRA-3* is a gene coding for a beta-lactamase originally identified in soil samples and could possibly contribute to imipenem resistance [26]. The gene, *erm(38)*, encodes for a 23S dimethyltransferase found in *Mycobacterium smegmatis* and can provide resistance to macrolides and lincosomides [27]. It has been shown to increase resistance to Clarithromycin [28]. mtrA is a gene from *Mycobacterium tuberculosis* whose expression has been shown to influence cell morphology and drug resistance in *Mycobacterium smegmatis* [29, 30].

We also used RGI to identify genes with lower homology to known resistance genes, providing possible starting points for follow up studies. We found a list of candidate genes that may contribute to resistance and further narrowed down the list using the susceptibility testing data by filtering out genes that were present in susceptible populations and genes that were not present in intermediate or fully resistant populations. Only 6 of the 11 drugs had low identity matches associated with their resistance pattern: TMP-SMX (sulfonamide), ciprofloxacin (fluoroquinolone), moxifloxacin (fluoroquinolone), imipenem (carbapenem), cefoxitin (cephalosporin), and Minocycline (tetracycline). Genes are listed in S4 Table.

We further examined genes reported to be targets of mutations that may cause resistance, such as mutations in 23S rRNA for linezolid resistance and gyrA for fluoroquinolone resistance. There were no known or novel mutations present in the 23S rRNA genes in our isolates

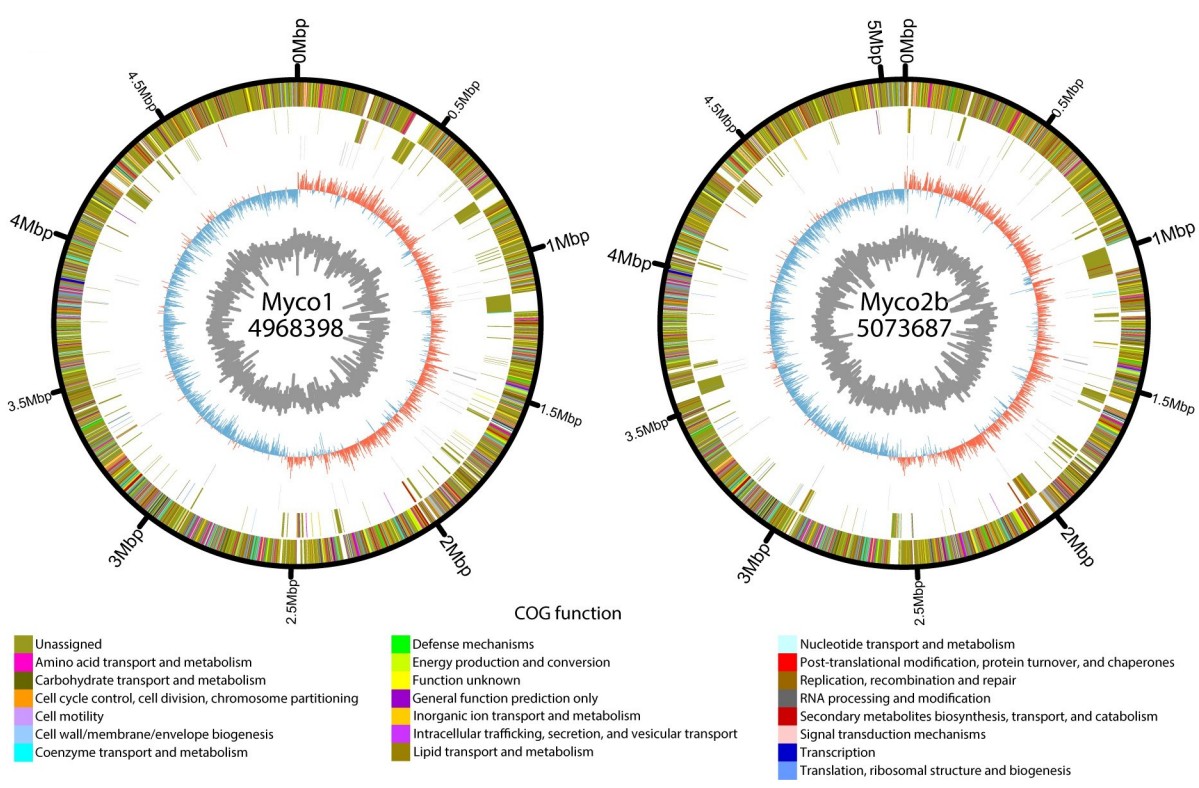

**Fig 3. Comparison of genomes from the candidate transmission pair.** Circos plots are shown of assemblies of Myco1 (left) and Myco2b (right). Each ring represents, from inner to outer ring GC content; GC skew; RNA genes; genes unique to the isolate, colored by COG function; and genes shared between the two isolates, colored by COG function. Each genome's start positionwas rotated to the origin of replication and genomes with multiple contigs were merged to form a single contig for visualization.

that correlated with drug susceptibility or resistance. For gyrA, we compared the two intermediate resistant isolates (Myco6 and Myco7) to five resistant isolates. We found that there were no consistent mutations that resulted in resistance, but Myco6 had two mutations (S710R and E773D) not present in the other isolates and Myco7 had one mutation not present in the other isolates (T832A).

## Discussion

Here we investigated the possibility of a nosocomial *M. chelonae* transmission at the HUP in 2017 and found that there is no evidence to support a point source outbreak. Our analysis found each isolate of *M. chelonae* was no more similar to each other than to the strains collected from the NCBI database. This supports that these infections were instead likely acquired from the environment and not from a single point source. Our analysis supports the use of next generation sequencing to assess possible nosocomial transmissions within the hospital environment.

Our data provide perspective on the amount of genetic variation in the population of *M. chelonae* in Philadelphia. The number of SNVs found in our isolates of *M. chelonae* was up to 62,854 SNVs (19,082.6 SNVs per Mpb). This is similar to *Mycobacterium marinum*, a species that targets fish, and *Mycobacterium avium*, a species that can cause respiratory illness in humans, which showed up to 89,000 SNVs (24,054.1 SNVs per Mbp) and 26,871 SNVs (6249.1 SNVs per Mbp), respectively. Both show more variation than *Mycobacterium tuberculosis*, which showed a maximum 1800 SNVs between strains [31–33].

To provide a control for sequencing error, we developed a dataset to establish an empirical threshold for calling identity between two isolates that takes account of error in DNA sequence determination. We sequenced a strain of *V. campbellii* using the Illumina method 39 times independently. The number of SNVs ranged from 0 to 74 SNVs depending on which pair of sequencing runs was compared, despite the same input DNA. There were several runs that, though following the same protocol, had higher numbers of SNVs on average in pairwise comparisons, and these correlated with low sequence coverage (S3 Table). Upon considering the length of the core genome, the average SNVs per Mbp was 3.6 which could be used for a general threshold for comparing microbial genomes considering sequencing error. Going forward, the data presented here provides useful background for comparing microbial genomes.

Our analysis also included two samples collected from the same individual and a single isolate sequenced twice, again to help evaluate thresholds for calling identity despite sequencing error. We found that our two samples from the same patient only differed by 0.91 SNVs per Mbp core genes after our hybrid assembly; our single isolate sequenced twice differed by 0.61 SNVs per Mbp core genes. These control *M. chelonae* genomes show SNV numbers well within the range expected for identical sequences based on our *V. campbellii* dataset. This indicates that the sequencing and hybrid assembly pipeline works as expected to generate high quality genomes and allows identification of identical organisms sequenced twice independently.

Our analysis also shed light on the host preferences of sequenced *M. chelonae* strains. Some of our isolates were closely related to environmental samples annotated as from water and sea creatures, while others clustered with human isolates. Our data did not provide evidence for a strongly human-associated clade.

Examination of SNVs in particular genes and RGI provided a list of possible mechanisms of resistance. For the SNVs that were identified in *gyrA*, the resulting amino acid changes might potentially modulate the resistance phenotype, though functional confirmation is needed. As for the genes from the RGI, while there were a few high identity resistance gene hits in our isolates, the presence of these genes did not correspond with the drug susceptibility phenotypes. For example, every isolate had an 80% identity hit to *erm(38)*, a gene found in *M. smegmatis* to mediate resistance to macrolides, such as clarithromycin, but all of the isolates were susceptible to clarithromycin. Thus, it is possible that the *erm(38)* homolog in *M. chelonae* performs a different role. For *LRA-3*, a metallo-beta-lactamase gene, all isolates were intermediately or fully resistant to the carbapenem tested. Since there are two phenotypes in the presence of the gene, this suggests that it may play a role in resistance, but other mechanisms may be involved as well. We also assessed low homology hits to resistance genes and mutations in genes that are known to influence drug susceptibility. For both, we have a list of possible genes or mutations, but given our low sample size, it is likely that many are false positives, and all require further validation to confirm. Resistance in *M. chelonae* may in part be due to blocking entry of antibiotics by the tough cell wall documented previously [34]. Data presented here may help guide future studies of mechanisms of resistance in *M. chelonae*.

In conclusion, we investigated a possible nosocomial outbreak of *M. chelonae* at HUP. Our analysis concluded that no point source transmission occurred and that each case of *M. chelonae* involved clearly distinct strains, likely acquired from the environment. Our analysis also includes a dataset to help determine thresholds for evaluating identity between different strains while controlling for sequencing error. Finally, we queried potential antibiotic resistance mechanisms by genomic analysis, providing candidate genes and mutations for potential follow up.

## Supporting information

**S1 Table. Genome sequences analyzed in this study.**
(XLSX)

**S2 Table. List of core genes for Myco3a and their single nucleotide variations.**
(XLSX)

**S3 Table. *Vibrio campbellii* assembly information.**
(XLSX)

**S4 Table. List of potential *Mycobacterium chelonae* resistance genes by drug type.**
(XLSX)

## Acknowledgments

We are grateful to members of the Bushman laboratory for help and suggestions; and Laurie Zimmerman for help with figures.

## Author Contributions

**Conceptualization:** Christopher H. Gu, Chunyu Zhao, Laurel Glaser, Robert Baldassano, Kyle Bittinger, Lisa M. Mattei, Frederic D. Bushman.

**Data curation:** Christopher H. Gu, Chunyu Zhao.

**Formal analysis:** Christopher H. Gu, Chunyu Zhao.

**Funding acquisition:** Robert Baldassano, Frederic D. Bushman.

**Investigation:** Christopher H. Gu, Chunyu Zhao, Casey Hofstaedter, Laurel Glaser, Lisa M. Mattei.

**Methodology:** Christopher H. Gu, Chunyu Zhao, Casey Hofstaedter, Laurel Glaser, Lisa M. Mattei.

**Project administration:** Robert Baldassano, Kyle Bittinger, Frederic D. Bushman.

**Resources:** Christopher H. Gu, Chunyu Zhao, Casey Hofstaedter, Pablo Tebas, Laurel Glaser, Kyle Bittinger, Lisa M. Mattei, Frederic D. Bushman.

**Software:** Christopher H. Gu, Chunyu Zhao.

**Supervision:** Robert Baldassano, Kyle Bittinger, Frederic D. Bushman.

**Validation:** Christopher H. Gu, Chunyu Zhao.

**Visualization:** Christopher H. Gu, Chunyu Zhao.

**Writing – original draft:** Christopher H. Gu, Frederic D. Bushman.

**Writing – review & editing:** Christopher H. Gu, Chunyu Zhao, Casey Hofstaedter, Pablo Tebas, Laurel Glaser, Robert Baldassano, Kyle Bittinger, Lisa M. Mattei, Frederic D. Bushman.

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
