## [Decision Letter · Decision Letter 0]

17 Aug 2020

PONE-D-20-20588

Investigating hospital *Mycobacterium chelonae* infection using whole genome sequencing and hybrid assembly

PLOS ONE

Dear Dr. Bushman,

Thank you for submitting your manuscript to PLOS ONE. After careful consideration, we feel that it has merit but does not fully meet PLOS ONE’s publication criteria as it currently stands. Therefore, we invite you to submit a revised version of the manuscript that addresses the points raised during the review process.

Both reviewers have made numerous useful suggetions which should be addressed during revision. The comments by Reviewer 1 are included in the attached PDF file as comments on your original submission. The comments by Reviewer 2 are found in the latter part of this letter. To expedite the review of your revised manuscript, please consider consolidating your responses to the reviewers' points wherever possible.

We look forward to receiving your revised manuscript.

Kind regards,

Herman Tse

Academic Editor

PLOS ONE

Journal Requirements:

2. Thank you for stating in the text of your manuscript "samples were collected as part of routine clinical practice with patient consent ". Please clarify whether consent was informed and what type of consent was obtained (i.e., written or verbal; if verbal, please indicate how consent was witnessed and recorded). Please add all of this information to your ethics statement in the online submission form.

3.PLOS ONE requires experimental methods to be described in enough detail to allow suitably skilled investigators to fully replicate and evaluate your study. See https://journals.plos.org/plosone/s/submission-guidelines#loc-materials-and-methods for more information.

To comply with PLOS ONE submission guidelines, in your Methods section, please provide details relating to your methodology, specifically:

-primers used for hsp65 gene sequencing and any citation related to the primers

-catalog numbers for  RAPMYCO microbroth dilution plate

-citation for CLSI M42 A2 methods

-the numerical breakpoints used for Tigecycline suseptibility, based on Wallace et al.

-accession numbers of all reference genomes and all sequences of the isolated samples from your hospital

-link or citation for the Sunbeam pipeline and Albacore

Reviewers' comments:

Reviewer's Responses to Questions

**Comments to the Author**

1. Is the manuscript technically sound, and do the data support the conclusions?

Reviewer #1: Partly

Reviewer #2: Yes

2. Has the statistical analysis been performed appropriately and rigorously? 

Reviewer #1: Yes

Reviewer #2: Yes

3. Have the authors made all data underlying the findings in their manuscript fully available?

Reviewer #1: Yes

Reviewer #2: Yes

4. Is the manuscript presented in an intelligible fashion and written in standard English?

Reviewer #1: Yes

Reviewer #2: Yes

5. Review Comments to the Author

Reviewer #1: Good evening Authors,

Thank you for allowing me the chance to review this scientific research. I believe you have some very interesting data here. The use of the V. campbellii isolate as a model to determine the viability of calling identity on fully sequenced and assembled isolates is really cool. However, I do believe that the paper needs a bit more work as far as organization and clarity. Also, I think that some important points about your isolates and the comparison with the reference genomes are missing in the results as well as the discussion. A lot of the discussion comes off as a reiteration of the results. Thus, a re-tooling of the discussion to show the relevance of your findings in the current medical/microbial world or how your finding relate to other studies may be worthwhile. Please see the attached pdf with comments regarding the manuscript. I hope you find them helpful and use them as suggestions to increase the readers understanding of your novel findings.

Reviewer #2: The current study investigates the possible nosocomial transmission of Mycobacterium chelonae strains based on seven patient isolates collected during the period 2017 from the Hospital of the University of Pennsylvania (HUP). For the comparative genomic approach, they have used high-quality hybrid assemblies (Oxford Nanopore long reads and the Illumina as a high-quality short read). To rule out the possibility of sequencing errors Vibrio campbellii dataset has used as a positive control and sequenced 39 times. Single nucleotide variant (SNV) analysis and phylogenetic tree based on core genes (SNVs) reveal infections are possibly transmitted via environment source or other niche but not due to the outbreak at the hospital. Antibiotic resistance-related homolog search and their putative candidate genes list analysis provides more insights into the emerging nontuberculosis mycobacteria research. The manuscript is very clear, the experiments have been conducted diligently with the appropriate controls and overall supportive conclusions.

Minor

1) In Figure 1 a-b) it would be more useful to list all the species names or strain names.

2) In Figure 2) the resolution of the figure should be improved. As of now, it looks blurry image with the text font.

3) In Figure 3) Circosplot, the genome size of Myco1 has shown as 4958837. But for the same, in Table S1) it is 4968398 bp update the right genome size. Represent the outer ring with the genome reference scale (0.5, 1, 1.5,… 5 Mb) that would help readers to identify a unique region precise position.

4) Table S1) Write all the species names, strain names, source names, and accession number of newly sequenced genomes (Myco1-7). Also, Update with the genome coverage information for the Oxford Nanopore and Illumina datasets.

5) Ln 70, delete "that"

6) Update software version numbers for all the tools. For example Prokka ver---, SamTools ver---, CheckM ver---

7) Ln 152, rewrite as " 100% conserved, likely due to the draft genomes".

8) Ln 167, rewrite "45 to 64" instead "46 to 64"

9) Table 2) Footnote: Also write complete abbreviations for R, S and I

10) Ln 194, alignment to a reference genome (Myco1? it has not stated which was the reference was used to map SNVs as a reference).

11) Ln 210, we used all 43 genomes (as of now at NCBI,47 genomes available) possibly mention the latest sequence retrieved date. For these 43 genomes, is it PROKKA annotation or the NCBI annotation that has been used?

12) Ln 264, fewer core genes (3143 vs 3368). Not discussed how similar is this phylogeny compared to the hybrid assemblies phylogeny (if you have the data)? Fig 1a

13) Ln 270, "Providing strong evidence that they are different strains". possibly with the ANI values, this sentence makes strong evidence supporting different strains.

14) Ln 356, a suggestion to delete "Our investigation of the genetic basis of M. chelonae drug resistance was inconclusive". Line 378-379, the statement provides candidate genes.

15) It should be noteworthy, to discuss SNVs to their genomic positions such as is it belong to the coding region or intergenic sequence. Also, How many genes and which genes are affected, and their functional role.

16) SNVs per MB site was discussed in this work. It should be appropriate to compare how this SNVs per MB is in other mycobacteria eg. NTMs or pathogenic mycobacteria.

6. PLOS authors have the option to publish the peer review history of their article (what does this mean?). If published, this will include your full peer review and any attached files.

Reviewer #1: No

Reviewer #2: No

---

## [Author Response · Author response to Decision Letter 0]

30 Sep 2020

We thank the reviewer and editors for their comments upon our submission. Our attached rebuttal letter under the label "Response to Reviewers" provides responses to the reviewers comments.

---

## [Editor Report · Decision Letter 1]

21 Oct 2020

Investigating hospital *Mycobacterium chelonae* infection using whole genome sequencing and hybrid assembly

PONE-D-20-20588R1

Dear Dr. Bushman,

We’re pleased to inform you that your manuscript has been judged scientifically suitable for publication and will be formally accepted for publication once it meets all outstanding technical requirements.

Kind regards,

Herman Tse

Academic Editor

PLOS ONE
---

## [Editor Report · Acceptance letter]

29 Oct 2020

PONE-D-20-20588R1 

Investigating hospital *Mycobacterium chelonae* infection using whole genome sequencing and hybrid assembly 

Dear Dr. Bushman:

I'm pleased to inform you that your manuscript has been deemed suitable for publication in PLOS ONE. Congratulations! Your manuscript is now with our production department. 

Kind regards, 

on behalf of

Dr. Herman Tse 

Academic Editor

PLOS ONE